# *Monilinia fructicola* Response to White Light

**DOI:** 10.3390/jof9100988

**Published:** 2023-10-04

**Authors:** Juan Diego Astacio, Eduardo Antonio Espeso, Paloma Melgarejo, Antonieta De Cal

**Affiliations:** 1Grupo de Hongos Fitopatógenos, Departamento de Protección Vegetal, Centro Nacional INIA-CSIC, 28040 Madrid, Spain; juan.astacio@inia.csic.es (J.D.A.); melgar@inia.csic.es (P.M.); 2Programa Biotecnología y Recursos Genéticos de Plantas y Microorganismos Asociados, ETSIA, Universidad Politécnica de Madrid, 28040 Madrid, Spain; 3Laboratorio de Biología Celular de Aspergillus, Departamento de Biología Celular y Molecular, Centro Investigaciones Biológicas Margarita Salas, CSIC (CIB-CSIC), 28040 Madrid, Spain; eespeso@cib.csic.es

**Keywords:** brown rot, *vivid1*, transcriptional regulation, sporulation, photoreception

## Abstract

Light represents a powerful signal for the regulation of virulence in many microbial pathogens. *Monilinia fructicola* is the most virulent species causing brown rot in stone fruit crops. To understand the influence of light on *M. fructicola*, we measured the effect of white light and photoperiods on the colonial growth and sporulation of the model *M. fructicola* strain 38C on solid cultures. Searches in the *M. fructicola* 38C genome predicted a complete set of genes coding for photoreceptors possibly involved in the perception of all ranges of wavelengths. Since white light had an obvious negative effect on vegetative growth and the asexual development of *M. fructicola* 38C on potato dextrose agar, we studied how light influences photoresponse genes in *M. fructicola* during early peach infection and in liquid culture. The transcriptomes were analyzed in “Red Jim” nectarines infected by *M. fructicola* 38C and subjected to light pulses for 5 min and 14 h after 24 h of incubation in darkness. Specific light-induced genes were identified. Among these, we confirmed in samples from infected fruit or synthetic media that blue light photoreceptor *vvd1* was among the highest expressed genes. An unknown gene, *far1*, coding for a small protein conserved in many families of Ascomycota phylum, was also highly induced by light. In contrast, a range of well-known photoreceptors displayed a low transcriptional response to light in *M. fructicola* from nectarines but not on the pathogen mycelium growing in liquid culture media for 6 days.

## 1. Introduction

Brown rot is a fungal disease affecting stone and pome fruit crops worldwide, caused mainly by three species of the genus *Monilinia* spp. [1], *Monilinia fructicola* (G. Winter), *Monilinia fructigena* (Honey) and *Monilinia laxa* (Aderhold and Ruhland). However, *M. fructicola* has a higher growth and aggressiveness than *M. laxa* and *M. fructigena,* which may be the cause of the displacement of both species on stone fruit orchards [2]. In our laboratory strain, 38C has been largely used as a model to understand peach infection by *M. fructicola* [3].

*M. fructicola*, belonging to the Sclerotiniaceae family, has a necrotrophic lifestyle as it colonizes plant cell tissue and causes cell death in order to obtain nutrients [1]. The *M. fructicola* infection process can be divided into three phases: pre-penetration, penetration and colonization. Infection begins when *M. fructicola* conidia germinate on the fruit surface to produce germ tubes and/or appressoria, which then penetrate through the fruit surface depending on the prevailing environmental conditions [3] and degree of maturity of the fruit [4]. Temperature affects the production of germ tubes or appressoria by *M. fructicola* after conidial germination [4]. *M. fructicola* produces the same number of germ tubes or appressoria at 4 °C but produces more germ tubes than appressoria at temperatures higher than 10 °C. Negligible or no germination of *M. fructicola* conidia occurs at 60% relative humidity (RH) [4]. However, the effect of light on these early pre-penetration stages of *M. fructicola* is unknown.

Light has been identified as another important environmental factor in the development of plant diseases [5,6], affecting both the host response and the pathogen virulence. Light drives adaptive responses in fungi, which actively sense their environment in order to activate protective mechanisms against light-associated stress and to regulate many facets of their development, like carotenogenesis and conidiation [7], secondary metabolism [8] and the balance between sexual and asexual development [9]. Behind this regulation of cell development, there is a sophisticated light signaling machinery which is composed of different apoproteins in association with chromophores [10]. These complexes, named photoreceptors, react to light of a determined wavelength through absorption of a photon that promotes structural changes in the chromophore, which, in turn, induce conformational changes in the apoprotein [11]. There are several families of photoreceptors depending on the chromophore they harbor and the wavelength they are sensitive to: near-UV/blue-light-sensing cryptochromes (CRYs, 350–500 nm), blue-light-sensing LOVs (light, oxygen, voltage) domain-containing proteins (450 nm), green light-sensing opsins (Ops, 540 nm), and phytochromes which detect red/far red ratio (650–780 nm) [11].

Recent genomic studies in *M. laxa* suggest that the photoreceptor’s arsenal [12] is similar to those described in the well-known pathogen *Botrytis cinerea* [13], which also belongs to the Sclerotiniaceae family. A unique ortholog for each photoreceptor with a highly conserved domain architecture was identified in *M. laxa* [12]. *M. laxa* has proven to be a light-responsive plant pathogen since different light conditions, intensities and photoperiods affect its transcriptional profile, phenotype, growth and conidiation rate [14,15]. Light also affects the development of brown rot caused by *M. laxa* on nectarines and cherries [15]. In this study, we focused on determining whether the model for *M. fructicola* infection in peaches, strain 38C, would be able to perceive the presence or absence of light by identifying the completeness of photoreceptors’ machinery, which genes are under light regulation when the fungus is cultured on fruit in early pre-penetration stages or synthetic media, and how exposure to light influences the growth of this fungal plant pathogen.

## 2. Materials and Methods

### 2.1. Monilinia Fructicola Strains

Single-spore isolate from *M. fructicola* 38C is stored at the culture collection of the Plant Protection Department of INIA-CSIC (Madrid, Spain). From the same collection, we chose strain 1C as a control on solid cultures. The genome from *M. fructicola* 38C is sequenced and available at NCBI under the accession number GCA_016906325.1 (BioProject PRJNA503180) [16]. Isolates were maintained as cultures on Potato Dextrose Agar (PDA Difco^TM^, Franklin Lakes, NJ, USA) in darkness at 4 °C for short-term storage and as a conidial suspension in 20% glycerol at −80 °C for long-term storage. For conidia production, isolates were cultured on PDA at 25 °C for 7 days in continuous darkness.

### 2.2. Effect of Different Lights on Mycelial Growth and Sporulation of M. fructicola on PDA

Effect of white light, generated using a set of four fluorescent bulbs of Osram36W/954 “daylight”, on the growth rate and conidia production was evaluated for *M. fructicola* strains 1C and 38C. These bulbs produce an irradiance of 110.13 W m^−2^, a luminous flux of 2850 lm, illuminance of 8700 lux and have a wavelength range between 300 and 700 nm, with a maximum around 550 nm (Appendix A). Petri dishes filled with PDA were inoculated in the center with a 15 μL droplet of a suspension containing 10^6^ conidia mL^−1^ and cultivated for 7 days in growth chambers at 22–25 °C under two different photoperiods: 8 h light/16 h darkness and 12 h light/12 h darkness. Distance between light bulbs and samples was set to 25 cm and control groups were set in continuous darkness. Two perpendicular diameters were measured starting on the third day of incubation until the end of the experiment, and daily growth speed (diameter in mm day^−1^) was calculated using regression analysis from the average measurements of each colony diameter. Total sporulation was calculated at the end of each assay and then referenced to each colony growth area. In order to do so, the surface of each plate was individually scraped after adding 5 mL of sterilized distilled water and filtered through glass wool to remove the mycelia. The number of conidia produced was counted using a hemocytometer. Data were expressed as total conidia divided by the colony area (cm^2^). Five technical replicates were used for each condition and the whole experiment was repeated twice.

### 2.3. Effect of Different Lights on Gene Expression Levels of M. fructicola in Potato Dextrose Broth

Influence of white light generated using a set of four fluorescent bulbs of Osram 58W/840 “Cool White” was also tested. These bulbs produce an irradiance of 165.55 W m^−2^, a luminous flux of 5200 lm, illuminance of 14,900 lux and have a wavelength range between 300 and 700 nm, with a maximum around 520 nm (Appendix A). Flasks containing 100 mL of Potato Dextrose Broth (PDB, Difco^TM^, Franklin Lakes, NJ, USA) were inoculated with conidia from *M. fructicola* 38C and grown in darkness at 25 °C for 6 days to allow for mycelial growth. Lights were turned on and samples were taken at 5 min, 1 h, 2.5 h, 5 h and 14 h of uninterrupted illumination. Distance between light bulbs and samples was set to 25 cm and control groups were set in continuous darkness. Mycelium was then collected by filtering the culture through miracloth and immediately frozen in liquid nitrogen. Flasks maintained in darkness were used as control. Three biological replicates were used in each condition and the complete experiment was repeated twice.

### 2.4. Effect of Different Lights on Nectarine Brown Rot Disease

Nectarines (var. Red Jim) from Ebro valley (Lleida, Spain) harvested at optimal commercial maturity in the first weeks of August were used. Surface of nectarines were disinfected using the hypochlorite procedure [17]. Afterwards, nectarines were prepared for infection experiments by being dried in a laminar flow cabinet and then each nectarine was inoculated with three drops of 15 μL of a suspension containing 10^6^ conidia mL^−1^ from *M. fructicola* 38C. Inoculated fruit were placed on sterilized cellulose alveoli inside disinfected plastic containers lined up with soaked paper to maintain humidity. Each container was individually sealed with plastic film and incubated in the dark for 24 h at 22 °C and 100% RH to allow conidial germination and germ tube formation [4]. Next, lights were turned on and fruit tissue samples with germinated conidia were taken at 5 min, 1 h, 2.5 h, 5 h and 14 h of continuous illumination. Inoculated fruit maintained in darkness were used as control. After each incubation time, samples of peel and pulp tissues of 1 cm diameter encompassing the inoculation sites were collected from three individual fruits (three biological replicates) per condition. All samples were frozen in liquid nitrogen before being kept at −80 °C until further transcriptional profiling analysis.

### 2.5. RNA Extraction

Total RNA of *M. fructicola* 38C was extracted from infected fruit tissues according to the rapid CTAB-based protocol [18] adapted to *Monilinia* spp. [12]. Total RNA was resuspended in RNase-free Mili-Q water and frozen immediately at −80 °C until further use.

When using mycelium from PDB as source for RNA extraction, a TRIreagent based method was used (Sigma–Aldrich, St Louis, MO, USA), as previously described [19]. Total RNA was resuspended in RNase-free MiliQ water and frozen immediately until further use. 

In both cases, RNA concentration and purity were measured using a NanoDrop 2000 spectrophotometer and RNA integrity was checked by 1.2% agarose gel electrophoresis. 

### 2.6. Identification of Photoreceptor Coding Genes in M. fructicola 38C Genome

The blast tool from NCBI Genomic workbench (https://www.ncbi.nlm.nih.gov/tools/gbench/ accessed on 13 December 2022) was used to identify a list of putative photoreceptor coding genes and light-response genes, described in *M. laxa* [12], in the available 38C genome and proteome prediction.

### 2.7. Transcriptomic Analysis

RNA samples from nectarines infected by *M. fructicola* 38C, from the same Osram 58W/840 “Cool White” as before and darkness at 5 min and 14 h post irradiance (pi), were sequenced using Illumina technology by Macrogen facilities. RawData files in .fastq format were generated containing 150pb reads. These reads were trimmed and mapped against available *Monilinia* spp. genomes using CLC Genomics Workbench 22.0.3 [20] software and differential expression analyses were carried out between the light and darkness in both time stamps. A transcript was considered to be differentially expressed when it had a logFC lower or higher than −2/2, respectively, and a Reads Per Kilobase Million (RPKM) value higher than 20 in at least one condition.

Gene Ontology (GO) and functional annotation analysis were performed using OmicsBox software 22.0.3 [21]. GO enrichment analyses were also carried out using the “Gene Set Test” tool in CLC Genomics Workbench (QIAGEN CLC Genomics Workbench 22.0.3 [20]. Filtering parameters to define a differentially expressed gene (DEG) were an RPKM value above 20 and a *p*-value < 0.05.

### 2.8. Gene Expression Analysis

Before cDNA synthesis, total RNA samples from 38C were treated with DNase I (Invitrogen, Carlsbad, CA, USA) following manufacturer’s instructions. cDNA was synthesized from 1 μg of DNase-treated RNA using SuperScript First-Strand Synthesis System for RT-PCR and oligo(dT) primer (Invitrogen, Carlsbad, CA, USA) following manufacturer’s indications. Real-Time PCR was performed with a 7500 Fast Real-Time PCR (Applied Biosystems, Foster City, CA, USA) using GoTaq qPCR Master Mix (Promega, Madison, WI, USA). Three technical replicates were analyzed for each biological replicate for both target and housekeeping genes, with a total volume of 20 μL, containing 10 μL of 2× GoTaq qPCR Master Mix, 7.8 μL of nuclease-free water, 300 nM of each primer and 100 ng of cDNA. The cycling program was 2 min at 95 °C, followed by 40 cycles of 15 s at 95 °C and 1 min at 60 °C. After the amplification reaction, a melt curve analysis was performed to check the specificity. Primers (oligonucleotides listed in Appendix A) for histone *H3* gene (Ml_histoneH3) (BK012065) were used from [15] Primers for *vivid1* (*vvd1*), *far1*, phytochrome 1 and 2 (*phy1* and *phy2*), opsine1 and 2 (*op1* and *op2*), cryptochrome 1 and 2 (*cry1and cry2*) and velvet 2 and 4 (*vel2* and *vel4*) genes (gene codes are in Table 1) were designed using Vector NTI (Thermo Fisher, Waltham, MA, USA) and MEGA X [22].

### 2.9. Statistical Analysis

Data were analyzed by ANOVA analysis. When F-test was significant at *p* ≤ 0.05, the means were compared using Tukey´s multiple test range.

Gene expression levels were calculated using the 2^−ΔΔCt^ method [23].

## 3. Results

### 3.1. Effect of Light on Growth Rate and Sporulation of M. fructicola

Strain 38C has previously been used to study the infection process on peaches by *M. fructicola*. Strain 38C was subjected to different conditions of light exposure to determine the possible effects of light on its colony phenotype. As a control, *M. fructicola* strain 1C was used for comparison. Strains 1C and 38C showed differences in colony development when exposed to light. Strain 38C grew faster than strain 1C in every condition tested (Figure 1).

Since photoperiods are important for fungal growth [12], in order to quantify the effect of white light or darkness on the growth and conidiation rates of *M. fructicola*, we tested two different photoperiods with 8 and 12 h of light each. The growth speed of both *M. fructicola* isolates was affected by the 12 h light photoperiod. The 8 h light photoperiod only had a significant effect on the growth rate of isolate 38C (Figure 2A,B). However, the effect of light on the sporulation of both isolates was different (Figure 2C,D). The sporulation of 38C was reduced by two photoperiods, with a further reduction at 12 h of light photoperiod. Meanwhile, light did not have any effect on the sporulation of 1C. These data indicated that our model strain 38C might differentially respond to light.

### 3.2. Light Sensing Machinery in M. fructicola 38C

Fourteen genes encoding putative light sensing and photoresponse related (photoreceptors) proteins (listed in Table 1) were identified in the *M. fructicola* 38C genome after BLASTP searches using the NCBI Genomic Workbench and available genomic data from *B. cinerea* strain B05.10 and *M. laxa* strain 8L, together with the predicted proteome of *M. fructicola* 38C. Deduced protein sequences showed shared and well-conserved domains with those of *B. cinerea* and *M. laxa* (Table 1 and Table 2). 

The predicted domain architecture of photoreceptors agreed with previous studies. Three red/far red ratio light-sensing photoreceptors known as phytochromes (*mfcphy1*, *mfcphy2* and *mfcphy3*) were found. These proteins contain a light-sensing PAS-GAF-PHY module, in addition to domains commonly found in hybrid histidine kinases (HKs). Rhodopsins are green-light-driven proton pumps integrated in the cell membrane and carrying a covalently bound all-trans-retinal as the chromophore, locating two of this type of photoreceptors in *M. fructicola* strain 38C, *mfcop1* and *mfcop2*. For blue-light-sensing, there are many unrelated proteins, commonly referred as LOVs, with an LOV (Light, Voltage, Oxygen) domain bounded to a flavin adenine dinucleotide (FAD/FMN). *M. fructicola* strain 38C had three putative photoreceptors from two distinct LOVs groups, namely L1, which are GATA-type zinc transcription factors; and L2, a group of proteins with a conserved LOV domain but lacking any effector domain. These putative photoreceptors were *mfcwcl1*, *mfcwcl2* and *mfcvvd1*, respectively. The last photoreceptor group found in the predicted proteome of strain 38C were 2 cryptochromes, *mfccry1* and *mfccry2*. These proteins were very similar to photolyases that have gained a signaling function. They sense near-UV/blue light with methenyltetrahydrofolate (MTHF) as the antenna pigment and FAD as the photocatalytical chromophore, both covalently bound to PHR- and FAD-binding domains. Another important component of the light sensing and signaling machinery is the *velvet* protein family, which links development and secondary metabolism with light. *mfcvel1*, *mfcvel2*, *mfcvel3* and *mfcvel4* were found in 38C (Figure 3). The Gene Ontology and Functional Annotation analysis confirmed the putative photoresponse function of the genes identified in the 38C predicted proteome (Table 2).

### 3.3. Transcriptional Profiles in the Light and Darkness in M. fructicola

Disease symptoms were not visible on fruit inoculated with 38C conidial suspension after the incubation and irradiation period, nor on the control group in continuous darkness. A comparative analysis was performed among RNAseq samples of infected fruit tissue by 38C after 5 min and 14 h of daylight irradiation and continuous darkness. The majority of transcripts that *M. fructicola* 38C expressed during the infection process, 6309 at 5 min of illumination and 4800 after 14 h, did not show any significant differences in expression levels regarding light conditions. There was another group of transcripts, 3612 at 5 min of illumination and 4927 after 14 h, which were not expressed by the fungus or had expression values under the cut off. However, there was a set of transcripts that changed their expression profile which was affected by light conditions, 165 at 5 min of illumination and 359 after 14 h (Table 3). 

Among the 165 genes from 38C that showed a modified expression after 5 min of 58 w “daylight” exposure, 126 of which were upregulated while the remaining 39 were downregulated. At 14 h of illumination, the number of upregulated transcripts was 188, a 49.2% increase, and the amount also grew to 171 for the downregulated gene class, which meant a 47.0% increase. Samples at 5 min and 14 h of illumination shared 34 upregulated and 9 downregulated transcripts (Appendix A). Raw reads corresponding to this RNAseq are available in PRJNA1011491.

Using CLC Genome Workbench (Qiagen), we performed a GO enrichment analysis among the DEGs in our RNAseq results for biological process, cell wall component and molecular function categories. In total, we found 165 different GO terms for the 5 min comparison and 125 for the 14 h comparison (Figure 4). The most overrepresented GO terms at 5 min and 14 h post irradiation were peptide metabolic process (14 hpi) and metabolic process (5 min pi) for the biological process category, cytoplasmic part (14 hpi) and membrane part (5 min pi) for the cellular component category, and oxidoreductase activity (14 hpi) and catalytic activity (5 min pi) for the molecular function category. A summary of the GO terms search in DEGs is shown in Appendix A. 

We detected a shift in the *M. fructicola* 38C strain’s transcriptional activity due to light exposure in the number of DEGs and which genes were transcribed. The GO terms dealing with the carbohydrate metabolic process, polysaccharide catabolic process, extracellular region and hydrolase activity of different substrates were also enriched among DEGs. This suggests that light might have a regulatory function in the production of carbohydrate active enzymes (CAZymes), some of which are related to the host cell wall degrading process acting, thus, as pathogenicity factors for *M. fructicola*.

### 3.4. Photoregulation of Photoresponse Genes in M. fructicola

Among DEGs in RNAseq, only one of the putative photoreceptor coding genes was present in 38C, *vvd1*. The remaining predicted photoreceptor genes were unaffected by light (Table 4) or showed very low expression values (measured as transcript per millions, TPMs). Searching for upregulated transcripts that could be related to photoresponse in 38C, we found the gene MFRU_021g00620.1, named from now on *far1*, which was significantly upregulated in light over darkness in both time stamps (Table 4). The Gene Ontology and functional annotation did not assign any function nor GO terms for this gene.

### 3.5. Far1, a Small Protein Coding Gene Upregulated during Light Exposure

FAR1 is a short protein (82aa) with an undefined function, lacking any known conserved domains or associated GO Terms. Putative homologous genes to *far1* were found in related fungal genomes. The blastp tool in UniProt [24] returned 250 hits, of which 248 belonged to the Ascomycota phylum and all of them belonging to the Pezizomycotina subdivision, with hits in six of twelve classes within this clade. The most abundant classes were Eurotiomycetes, with 91 hits including genus *Aspergillus* and *Penicilium* spp.; Sordariomycetes, with 70 hits including genus like *Colletotrichum*, *Fusarium*, *Verticilium*, *Neurospora*, *Magnaporthe* and *Rosellinia* spp.; and Leotiomycetes, with 58 hits including genus like *Botrytis*, *Monilinia* and *Sclerotinia* spp. Thus, FAR1 is a widely distributed protein in the phylum Ascomycota, present in some important species of phytopathogenic fungi and well-studied model organisms (Appendix A).

When reconstructing the evolutionary relations using the amino acidic sequence of the protein, we found that all *Monilinia* spp. Sequences formed a clade together, with *Sclerotinia* spp. And *Botrytis* spp. As close relatives. The rest of the clades in the tree are much more uncertain, suggesting conserved functions in those orthologs belonging to the Leotiomycetes class (Figure 5).

### 3.6. Differential Expression Analysis Using RT-qPCR in M. fructicola

*far1* and only *vvd1* among the putative photoreceptors and putative light-sensing protein coding genes were differentially expressed under light in our transcriptomic analysis. We confirmed these transcriptional profiles using RT-qPCR. We used the same infected plant tissue samples by 38C that were the subjects of RNAseq analyses. Genes *vvd1* and *far1* showed statistically significant upregulation when exposed to 58 w daylight by RT-qPCR. *vvd1* greatly increased its expression levels after a short light pulse of 5 min, decreasing with time but still being significantly different than continuous darkness, and rising up again at the 14 h post irradiation time stamp. The expression levels of *far1* were upregulated upon light exposure, reaching its peak at 14 h post irradiance when 38C grew over plant tissue (Figure 6).

We also questioned whether the presence of fruit would modify the effect of light exposure on the expression levels of photoreceptors. Only 5 out of 8 putative photoreceptor genes and the unknown function gene *far1* significantly increased their expression levels over the time *M. fructicola* 38C was exposed to light when growing in PDB (Figure 7). Blue-light-sensing *vvd1* was the most upregulated gene, increasing its expression levels sixfold at 1 h post irradiation and maintaining these expression levels throughout the whole experiment. Genes encoding near-UV/blue-light-sensing proteins *cry1* and *cry2* both showed the same pattern, continuously increasing their expression levels but never reaching those of *vvd1*. *phy2*, encoding a red light photoreceptor, shares the same expression profile as *cry1* and *cry2* but with a lesser fold change overall. In contrast, green light photoreceptor *op2* activates its expression at five minutes of light exposure and reaches its highest at 1 h post illumination, at 3.5 fold, and then decreases its expression levels to not be significantly different from continuous darkness. Not all of these genes were modified in a continuous darkness condition (Figure 7). The remaining photoreceptor coding genes that were tested did not show any significant changes in gene expression under these conditions.

## 4. Discussion

White light emitted by the fluorescent bulbs used in this work has a significant effect on the growth and sporulation of *M. fructicola* and on some of its photoresponse genes in pre-penetration stages. Light is an essential source of abiotic environmental information for many fungi, regulating key elements of their behavior. *Aspergillus nidulans* is regulated in a light-dependent manner [26], with red light playing a dominant role, while *B. cinerea* needs light-dark cycles for the production of conidia [27]. The growth and conidiation of *M. laxa* on PDA were significantly affected by light and photoperiod [12]. However, *M. fructicola* responds to light stimulus differently than *M. laxa*, especially because the sporulation of *M. laxa* was always favored with light and photoperiod [12,28], an effect not observed in *M. fructicola*. 

The light effect on fungal growth and sporulation might be specific to fungal genus, species of the same genus, strains or different light sources [26,29]. The same isolate of *M. fructicola* 38C showed no difference in growth and sporulation under LEDs and darkness [30], which could be explained by the different qualities of light emitted by LEDs vs. fluorescent tubes. Also, there are differences in light wave composition between different models and brands of fluorescent light bulbs. For example, our light sources had less quantity of light emitted in the 400–450 nm range than the 500–550 nm range, which could affect the expression of genes controlled by photoreceptors sensing light of either of those wavelengths. This could explain why we were unable to find photoactivation of the white collar complex (WCC) as this photoreceptor senses light with a wavelength of 450 nm. 

Furthermore, growth and sporulation differences between the two *M. fructicola* strains were observed in the present study. While strain 1C, less virulent than 38C, did not show any statistically significant difference in conidia production between light and continuous darkness, strain 38C was unable to produce conidia under the two photoperiods tested and only produced conidia under continuous darkness. There are reported cases in the literature of “blind” strains resulting in anomalous regulation of conidia production in which the mutation resulting in such phenotype has been identified in a photoreceptor or a light-response-related protein. *Botrytis cinerea* T4 is a wild isolate with a mutation in the Velvet protein BcVEL1. It produces a truncated protein resulting in a loss of function. This strain is considered as “blind” and shows an “always conidia” phenotype, producing conidia regardless of illumination conditions. This phenotype is also associated with a reduction in virulence on several plant hosts [31]. This shows that *vel1* might be implicated in the regulation mechanisms for conidia production. The “always conidia” phenotype is also found in *A. nidulans* carrying the *veA1* mutation [26]. *veA* is a homologous gene to *vel1* from *B. cinerea*. The *A. nidulans* VeA1 mutant protein lacks the first 36 amino acids at the N-terminus and is not able to migrate from the cytoplasm to the nucleus, failing to respond to light, thus explaining the light independent conidiation phenotype displayed by those strains carrying the *veA1* allele [32]. In *B. cinerea*, a mutation in the *BcveA* gene and loss-of-function mutants of the photoreceptor coding gene *bcphy3* were identified, all causing evident morphological defects and reduced virulence in a light-independent way [33]. However, even though *M. fructicola* is more virulent than *M. laxa*, no differences were found between its photoreceptors that could explain the light effect on 38C conidia production.

Differential transcriptional regulation mediated by light of photoreceptors and light-response-related proteins could also explain the anomalous conidiation phenotype of strain 38C. Some transcripts expressed by *M. fructicola* 38C during the fruit infection process and measured by RNAseq showed significant differences between light and dark infection conditions in the present study. Among photoreceptors, only *vvd1* showed significant differences in its transcripts between light and dark infection conditions in infecting plant tissue. However, these results could be affected by growth culture media and, as we discussed before, the light wave composition of light sources. Light had a significant effect on the *M. fructicola* 38C strain’s transcriptional profile, affecting the genetic expression of key photoreceptors, namely *vvd1*, *cry1*, *cry2*, *op2* and *phy2*, when *M. fructicola* was cultured in liquid medium. All the above photoreceptors were also found to be light inducible in *M. laxa* strain 8L [12]. They also found that light produces differential expressions on *vel1* and *vel4*, which are involved in conidiation regulation, but we found no light effect on 38C for those genes. However, *M. fructicola* 38C showed serious difficulties to sporulate on PDA after incubation for 6 days under white light.

While we did not find any evidence of photo regulation for *wcl1* and *wcl2* among the DEGs in our RNAseq on infected nectarine tissue, our study shows that *vvd1*, a blue light photoreceptor with an antagonistic regulatory function of the WCC by physical interaction [34], had an early, high and consistent photoinduction in every condition we tested. Another photoreceptor that could be involved in explaining our results was *cry2*, which had a role in the negative regulation of conidia formation [13] and was found to be photoinducible for *M. fructicola* strain 38C in PDB. Additional studies are required to determine whether *vvd1* or *cry2* have a role in regulating conidiation in *M. fructicola* and if it could explain 38C’s anomalous conidiation behavior. *Wcl1* and *wcl2* join together to form the White Collar Complex (WCC), which is a primary regulator of light-responsive genes [7,35,36]. These photoreceptors did not show any light regulation in *M. laxa* [12] and some unpublished results suggest the same for *M. fructigena*.

On the other hand, gene MFRU_021g00620.1, from now on named *far1*, was significantly upregulated in light against darkness on our RNAseq on infected plant tissue. And, RT-qPCR confirmed these results using total RNA from mycelial samples grown in liquid culture. *far1*, which encodes for a small protein and which may serve as a light response signal, is present in some *Monilinia* species and other related fungi. The Far1 protein does not have any known function and our GO annotation did not assign any GO terms; however, a putative homologue of Far1 in *B. cinerea* is already labeled as “conidiation protein” in the Uniprot data base. Far1 is highly induced by light and might also play a role in explaining the abnormal conidiation behavior of 38C.

Further investigation is required to unravel the function of *far1* and its role in photoreception. Genetic transformation is a powerful tool for functional characterization of genes; however, *M. fructicola* strain 38C is recalcitrant to transformation and, due to the lack of successful and consistent transformation protocols, we have not been able to perform gene knockout or overexpression experiments to confirm a possible direct function of Far1 in photoreception. Our laboratory has made some efforts in order to overcome this situation but success in obtaining mutant strains is daunting [37]. Both *vvd1* and *far1* showed early, high and consistent light upregulation in all the conditions tested in this study and are present in all *Monilinia* species. Thus, we think they are perfect candidates for light response signaling genes.

## Figures and Tables

**Figure 1 jof-09-00988-f001:**
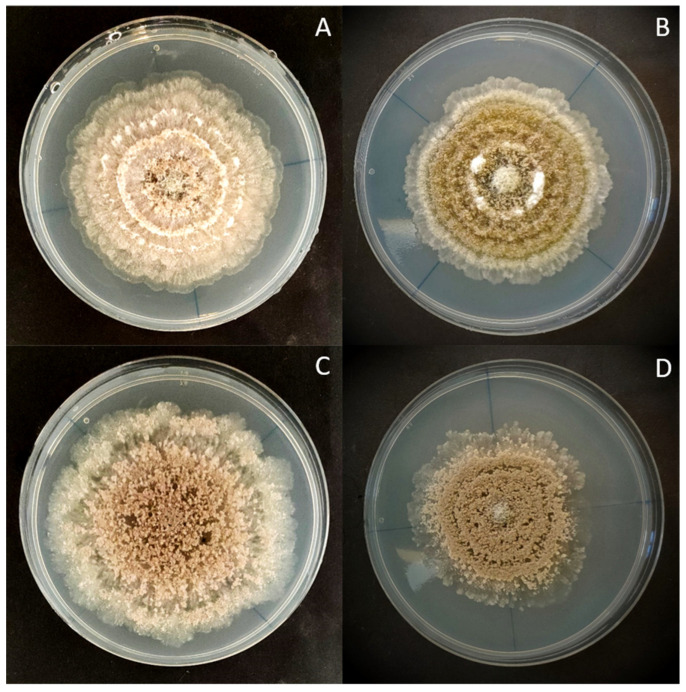
Effect of white light on two *Monilinia fructicola* strains after seven days of incubation on Potato Dextrose Agar (PDA): (**A**) strain 38C exposed to a 12 h white light photoperiod; (**B**) strain 1C exposed to a 12 h white light photoperiod; (**C**) strain 38C exposed to continuous darkness; (**D**) strain 1C exposed to continuous darkness.

**Figure 2 jof-09-00988-f002:**
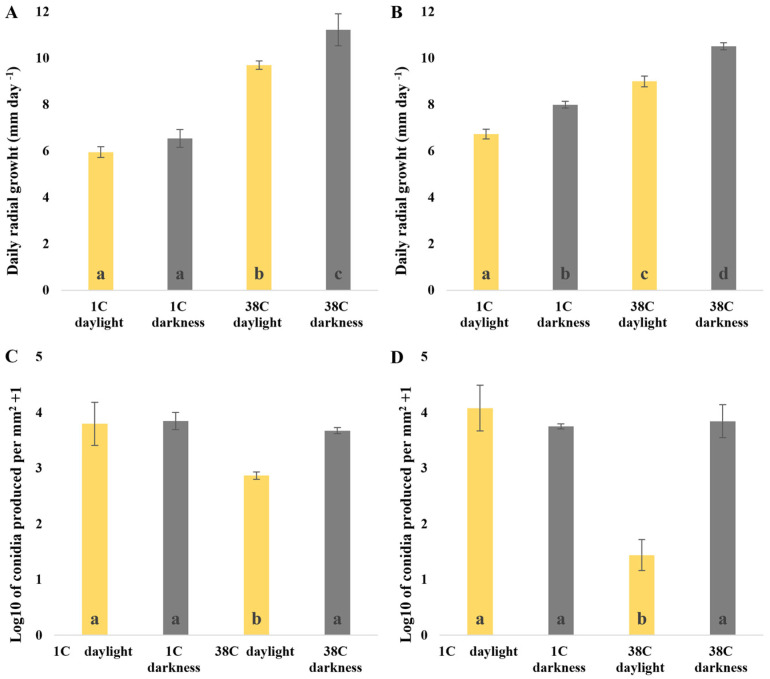
Effect of two different white light photoperiods on mycelial growth rate and sporulation of two strains of *M. fructicola*, 38C and 1C, after seven days of incubation on Potato Dextrose Agar (PDA): (**A**) mycelial growth rate under 8 h light photoperiod; (**B**) mycelial growth rate under 12 h light photoperiod; (**C**) sporulation under 8 h light photoperiod; (**D**) sporulation under 12 h light photoperiod. Data are the average and standard deviation from two completed assays with five technical replicates each one. Bars with same letter in each graph were not significantly different by Tukey’s multiple test range.

**Figure 3 jof-09-00988-f003:**
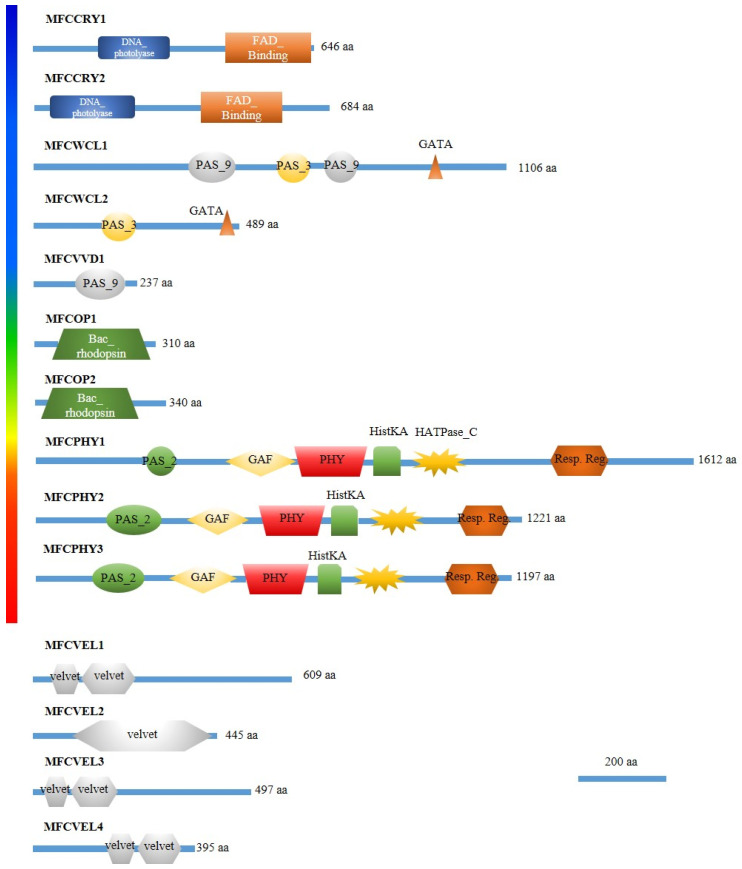
Domain architecture of putative photoreceptors and velvet family proteins in *Monilinia fructicola* strain 38C. DNA_photolyase (PF00875), FAD_binding (PF03441), PAS_9 (PF13426), PAS_3 (PF08447), GATA (PF00320), Bac_rhodopsin (PF01036), PAS_2, (PF08446), GAF (PF01590), PHY (PF00360), HisKA (PF00512), HATPase_C (PF02518), Response_reg (PF00072), Velvet (PF11754).

**Figure 4 jof-09-00988-f004:**
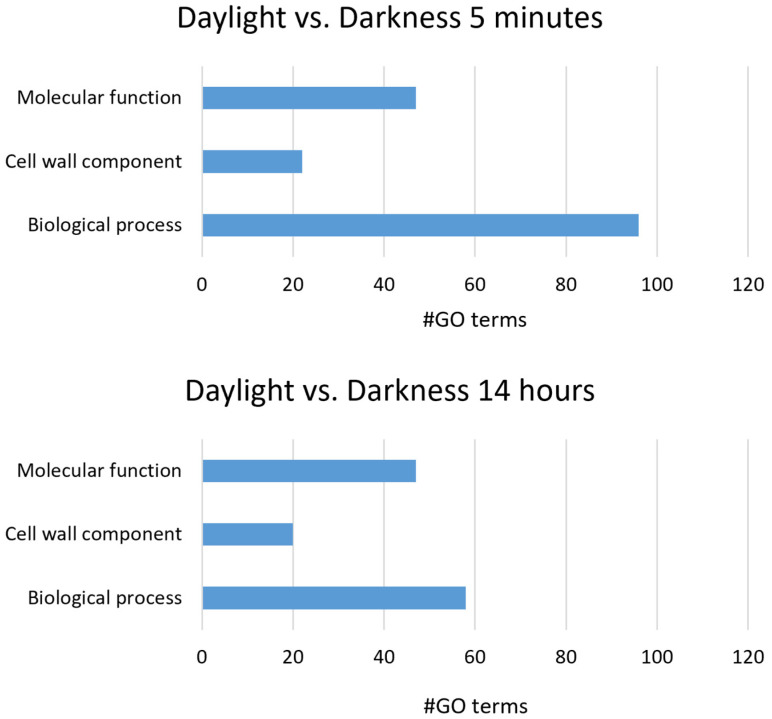
Distribution of GO terms by category on the Differentially Expressed Genes (DEGs) at 5 min and 14 h post irradiation.

**Figure 5 jof-09-00988-f005:**
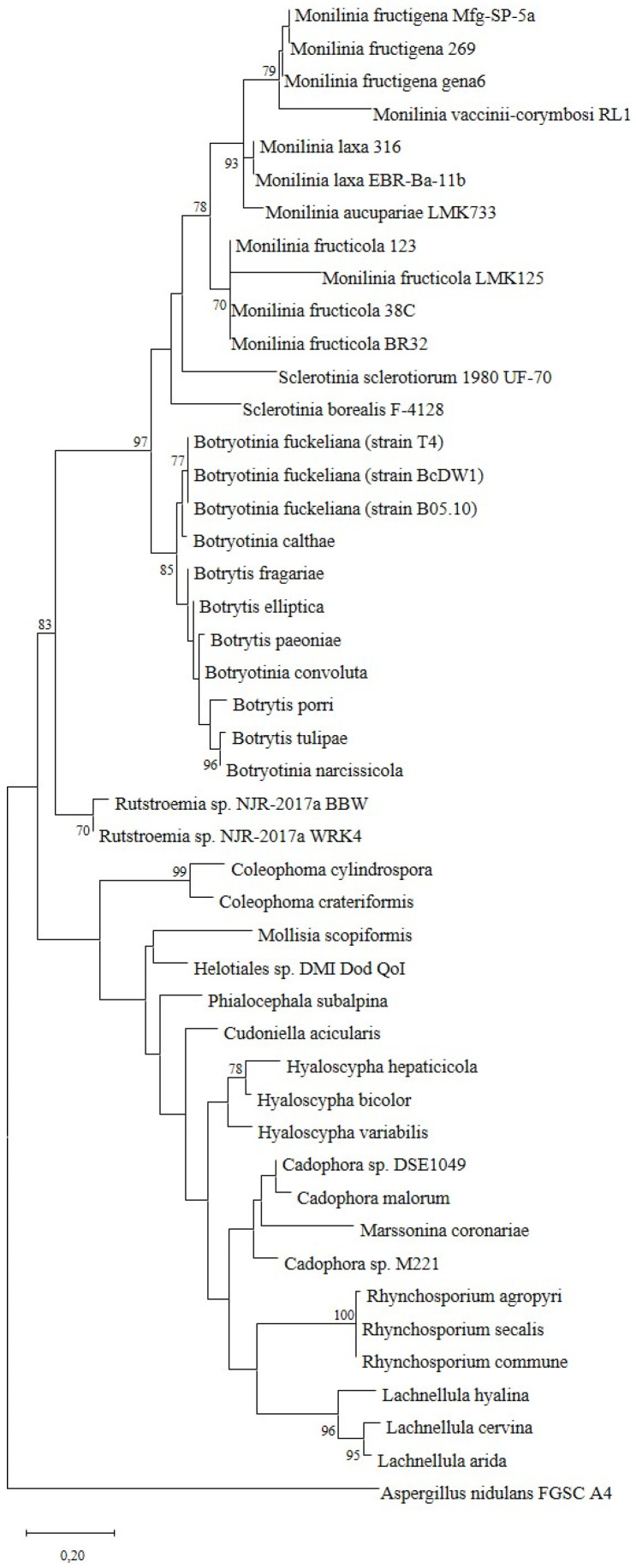
Evolutionary analysis of *far1* orthologs by Maximum Likelihood method The evolutionary history was inferred by using the Maximum Likelihood method and JTT matrix-based model [25]. The tree with the highest log likelihood (−2766.09) is shown. The percentage of trees in which the associated taxa clustered together is shown next to the branches. Initial tree(s) for the heuristic search were obtained automatically by applying Neighbor-Join and BioNJ algorithms to a matrix of pairwise distances estimated using the JTT model, and then selecting the topology with superior log likelihood value. The tree is drawn to scale, with branch lengths measured in the number of substitutions per site. This analysis involved 46 amino acid sequences. There was a total of 221 positions in the final dataset. Evolutionary analyses were conducted in MEGA X [22].

**Figure 6 jof-09-00988-f006:**
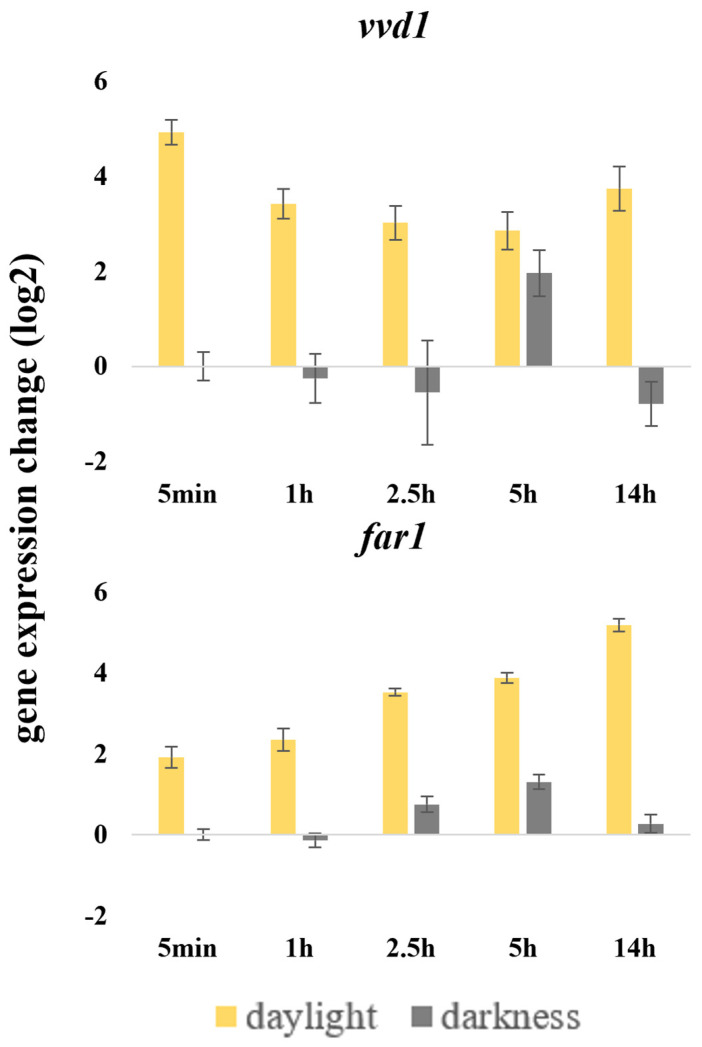
Changes in relative gene expression of two putative light-sensing protein coding genes (*vvd1* and *far1*) after daylight exposure and continuous darkness at different time points of *M. fructicola* strain 38C growing on fruit tissue. The scale is Log_2_ 2^−∆∆Ct^ of mean fold change values, including standard deviation, from three biological replicates, with three technical replicates each, after normalization against 0 h post illumination using the 2^−∆∆Ct^ method [23], with histone H3 as endogenous control gene.

**Figure 7 jof-09-00988-f007:**
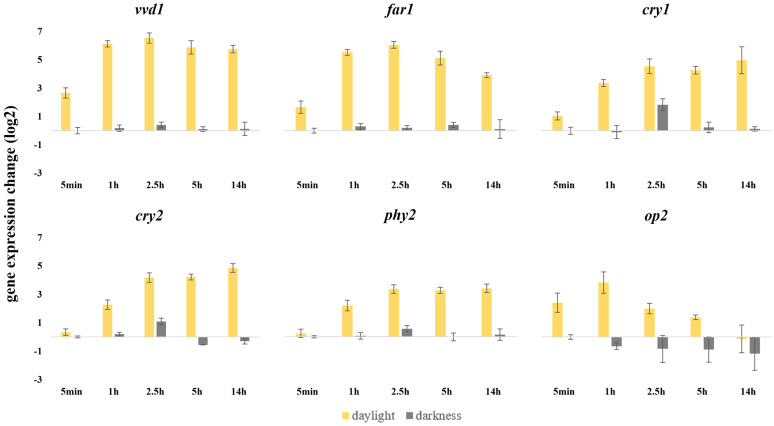
Changes in relative gene expression of six putative light-sensing protein coding genes after daylight exposure and continuous darkness on different time stamps when *M. fructicola* strain 38C grew in Potato Dextrose Broth (PDB). The scale is Log_2_ 2^−∆∆Ct^ of mean fold change values from two separate experiments, including standard deviation, from three biological replicates, with three technical replicates each, after normalization against 0 h post illumination using the 2^−∆∆Ct^ method [23], with histone H3 as endogenous control gene.

**Table 1 jof-09-00988-t001:** Photoreception related proteins presented in *M. fructicola* 38C and homology of their conserved domains in *Botrytis cinerea* and *Monilinia laxa*. ^a^ Protein unique identifier in 38C predicted proteome. ^b^ Putative protein function based on functional annotation. ^c^ Predicted protein length in 38C proteome ^d^ Gene unique identifier in *M. laxa* 8L genome; BLASTP identity and coverage. ^e^ Gene unique identifier in *B. cinerea* B05.10 genome; BLASTP coverage and BLASTP identity.

Gene Name	Seq ID ^a^	Putative Protein Function ^b^	Predicted Protein Length ^c^	*M. laxa* 8L ID ^d^	% Identity	% Coverage	*B. cinerea* B05.10 Id ^e^	% Identity	% Coverage
Putative Near-UV/Blue Light Sensors									
*mfccry1*	MFRU_030g01000	putative deoxyribodipyrimidine photo-lyase	646	*mlcry1*	92.01	89.54	*bccry1*	77.28	99.15
*mfccry2*	MFRU_004g02180	DASH family cryptochrome protein	684	*mlcry2*	94.6	100	*bccry2*	84.36	100
Blue light sensing									
*mfcwc1*	MFRU_072g00010	putative white collar-1 protein	1106	*mlwcl1*	96.26	99.91	*bcwcl1*	67.38	97
*mfcwl2*	MFRU_002g04340	putative white collar-2 protein	483	*mlwc2*	95.65	100	*bcwcl2*	72.27	99
*mfcvvd1*	MFRU_006g01210	vivid PAS VVD protein	237	*mlvvd1*	94.09	100	*bcvvd1*	73.84	100
Green light sensing									
*mfcop1*	MFRU_001g04210	putative opsin-1 protein, translocase	310	*mlops1*	96.12	100	*bcpop1*	87.5	100
*mfcop2*	MFRU_009g00610	putative opsin-like protein, translocase	340	*mlops2*	89.12	100	*bcpop2*	81.47	100
Red/far red ratio sensing									
*mfcphy1*	MFRU_022g00870	PHY1, histidine kinase-group VIII protein	1612	*mlphy1*	94.18	99.75	*bcphy1*	77.91	99
*mfcphy2*	MFRU_023g00600	PHY2, histidine kinase-group VIII protein	1221	*mlphy2*	92.08	99.59	*bcphy2*	75.69	100
*mfcphy3*	MFRU_005g01680	PHY3, histidine kinase-group VIII protein	1197	*mlphy3*	93.98	100	*bcphy3*	76.11	97
**Implicated in photoresponse**									
*mfcvel1*	MFRU_013g00210	velvet complex subunit 1	609	*mlvel1*	94.39	100	*bcvel1*	72.76	96
*mfcvel2*	MFRU_014g01690	velvet complex subunit 2	445	*mlvel2*	95.74	94.61	*bcvel2*	86.92	94
*mfcvel3*	MFRU_027g01050	velvet 3	497	*mlvel3*	95.57	100	*bcvel3*	79.84	100
*mfcvel4*	MFRU_018g00640	velvet 4	395	*mlvel4*	95.16	98.4	*bcvel4*	87.11	95

**Table 2 jof-09-00988-t002:** Gene ontology (GO) and functional annotation for putative photoreception related genes identified in the *M. fructicola* 38C predicted proteome.

Gene	Protein ID	GO ID	GO Name	Enzyme Name
*vvd1*	MFRU_006g01210	C:GO:0005634	C:nucleus	putative vivid pas protein
*cry1*	MFRU_030g01000	P:GO:0000719; P:GO:0018298; F:GO:0016829; F:GO:0097159; F:GO:1901363	P:photoreactive repair; P:obsolete protein-chromophore linkage; F:lyase activity; F:organic cyclic compound binding; F:heterocyclic compound binding	Lyases
*cry2*	MFRU_004g02180	P:GO:0006281; P:GO:0018298; P:GO:0060258; P:GO:0075308; F:GO:0003913	P:DNA repair; P:obsolete protein-chromophore linkage; P:negative regulation of filamentous growth; P:negative regulation of conidium formation; F:DNA photolyase activity	Carbon-carbon lyases
*phy1*	MFRU_022g00870	P:GO:0000160; P:GO:0006355; P:GO:0009584; P:GO:0016310; P:GO:0018298; P:GO:1902531; F:GO:0000155; F:GO:0005524; F:GO:0009881; C:GO:0005737	P:phosphorelay signal transduction system; P:regulation of DNA-templated transcription; P:detection of visible light; P:phosphorylation; P:obsolete protein-chromophore linkage; P:regulation of intracellular signal transduction; F:phosphorelay sensor kinase activity; F:ATP binding; F:photoreceptor activity; C:cytoplasm	Transferring phosphorus-containing groups; Histidine kinase
*phy2*	MFRU_023g00600	P:GO:0000160; P:GO:0006355; P:GO:0009584; P:GO:0016310; P:GO:0018298; P:GO:1902531; F:GO:0000155; F:GO:0005524; F:GO:0009881; C:GO:0005737	P:phosphorelay signal transduction system; P:regulation of DNA-templated transcription; P:detection of visible light; P:phosphorylation; P:obsolete protein-chromophore linkage; P:regulation of intracellular signal transduction; F:phosphorelay sensor kinase activity; F:ATP binding; F:photoreceptor activity; C:cytoplasm	Transferring phosphorus-containing groups; Histidine kinase
*phy3*	MFRU_005g01680	P:GO:0000160; P:GO:0006355; P:GO:0009584; P:GO:0016310; P:GO:0018298; P:GO:1902531; F:GO:0000155; F:GO:0005524; F:GO:0009881; C:GO:0005737	P:phosphorelay signal transduction system; P:regulation of DNA-templated transcription; P:detection of visible light; P:phosphorylation; P:obsolete protein-chromophore linkage; P:regulation of intracellular signal transduction; F:phosphorelay sensor kinase activity; F:ATP binding; F:photoreceptor activity; C:cytoplasm	Transferring phosphorus-containing groups; Histidine kinase
*op1*	MFRU_001g04210	P:GO:0007602; P:GO:0018298; P:GO:0034220; F:GO:0005216; F:GO:0009881; C:GO:0016021	P:phototransduction; P:obsolete protein-chromophore linkage; P:ion transmembrane transport; F:ion channel activity; F:photoreceptor activity; C:integral component of membrane	Translocases
*op2*	MFRU_009g00610	P:GO:0007602; P:GO:0018298; P:GO:0034220; F:GO:0005216; F:GO:0009881; C:GO:0016021	P:phototransduction; P:obsolete protein-chromophore linkage; P:ion transmembrane transport; F:ion channel activity; F:photoreceptor activity; C:integral component of membrane	Translocases
*wcl1*	MFRU_072g00010	P:GO:0006355; F:GO:0008270; F:GO:0043565	P:regulation of DNA-templated transcription; F:zinc ion binding; F:sequence-specific DNA binding	putative white collar 1 protein
*wcl2*	MFRU_002g04340	P:GO:0006355; F:GO:0008270; F:GO:0043565; C:GO:0005634	P:regulation of DNA-templated transcription; F:zinc ion binding; F:sequence-specific DNA binding; C:nucleus	putative white collar-2 protein
*vel1*	MFRU_013g00210	P:GO:0030435; C:GO:0005634; C:GO:0005737; C:GO:0016021	P:sporulation resulting in formation of a cellular spore; C:nucleus; C:cytoplasm; C:integral component of membrane	Velvet 1
*vel2*	MFRU_014g01690	P:GO:0030435; C:GO:0005634; C:GO:0005737	P:sporulation resulting in formation of a cellular spore; C:nucleus; C:cytoplasm	putative developmental regulator protein
*vel3*	MFRU_027g01050	C:GO:0005634	C:nucleus	putative velvet 3 protein
*vel4*	MFRU_018g00640	C:GO:0005634	C:nucleus	putative vea protein

**Table 3 jof-09-00988-t003:** Summary of RNAseq results.

RNAseq Analysis	# Upregulated Transcripts	# Downregulated Transcripts	# Transcripts with No Changes	# No Expression Transcripts
5 min daylight against continuous darkness	126	39	6309	3612
14 h daylight against continuous darkness	188	171	4800	4927

**Table 4 jof-09-00988-t004:** Expression values measured in Transcripts Per Million (TPMs) for putative photoreception related genes in *M. fructicola* 38C.

Gene	5 min			14 h		
	TPM Daylight	TPM Darkness	Log_2_ FoldChange	TPM Daylight	TPM Darkness	Log_2_ FoldChange
*mfcfar1*	657.9	34.1	4.3	534.4	38.2	3.6
*mfccry1*	90.5	40.2	1.2	48.4	22.8	0.9
*mfccry2*	197.5	109.6	0.9	32	26	−0.5
*mfcwc1*	31.8	47.2	−0.5	24.4	21.7	0
*mfcwc2*	64.4	41	0.7	35.6	11.9	1.4
*mfcvvd1*	346.4	27.2	3.7	105.93	35.1	1.4
*mfcop1*	593.3	254.9	1.2	181.1	210.2	−0.4
*mfcop2*	13.2	1.4	3.3	3.5	4.5	0.2
*mfcphy1*	25.2	33.4	−0.4	11.1	15.6	1.2
*mfcphy2*	66.6	28.1	1.3	33.2	11.6	1.3
*mfcphy3*	0.7	0.2	1.5	0	0.5	−3.3
*mfcvel1*	99.9	141.7	−0.5	46.5	57.2	−0.5
*mfcvel2*	118.7	93.5	0.4	78.6	28.8	1.3
*mfcvel3*	4	5.1	−0.4	1.2	0	3
*mfcvel4*	20.3	23.6	−0.2	11.6	17	−0.7

## Data Availability

Data are available in a publicly accessible repository that does not issue DOIs. Publicly available datasets were analyzed in this study. These data can be found here: PRJNA1011491.

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
