# Peer review of "Monilinia fructicola Response to White Light"

_jof, 2023, doi:10.3390/jof9100988_

Round 1

Reviewer 1 Report (Previous Reviewer 2)

Dear authors,

I think the quality and especially the information regarding the used light sources significantly improved. This will increase the likelihood that the results can be repeated by others. There still is a missing information about the light intensity (How many lamps were used at 25cm distance?). 1 or many? This would change the light intensity even if the intensity for 1 lamp and the light spectrum is now shown in the manuscript.

I think it is fine but would suggest a final read over to make sure there are no bigger mistakes.  I think there is somewhere peek instead of peak,... such minor mistakes should be checked again.  

Author Response

Dear Reviewer,

We wanted to express our sincere gratitude for your continuous engagement and feedback, which helped to improve the clarity and readability of our manuscript titled "Monilinia fructicola response to white light", submitted to Journal of Fungi.

We have incorporated the solicited information about light sources used in the experiments described in the manuscript, and carefully revised wording and the use of English in line with your recommendations.

We are confident that the manuscript now adheres to the standards of publication in Journal of Fungi.

Once again, I extend my gratitude for your dedication to the review process and for your detailed feedback. Your expertise has been instrumental in the refinement of this work.

Thank you for your time and consideration.

Sincerely,

Corresponding author.

Dr. Antonieta De Cal

Reviewer 2 Report (New Reviewer)

The authors should check the use of capital letters and italics when referring to genes. Such as 22 or 27.

Line 31-32: Acronyms/Abbreviations/Initialisms should be defined the first time they appear in each of three sections: the abstract; the main text; the first figure or table. When defined for the first time, the acronym/abbreviation/initialism should be added in parentheses after the written-out form.

Line 47: When defined for the first time, the acronym/abbreviation/initialism should be added in parentheses after the written-out form. (RH)

Line 61-63: Space required before SI unit. (nm)

Line 66: Acronyms/Abbreviations/Initialisms should be defined the first time they appear in each of three sections: the abstract; the main text; the first figure or table. When defined for the first time, the acronym/abbreviation/initialism should be added in parentheses after the written-out form. (B. cinerea change to Botrytis cinerea). Such as Line 246,249,460

Line 95: ml change to mL

Line 94: Space required before SI unit. (μL)

Line 96: Space required before SI unit. (h) Such as Line 128, 201,217,218

Line 111: Space required before SI unit. (nm)

Line 114-115: 2.5 h 5 h change to 2.5 h, 5 h

Line 125: ml change to mL

Line 129: HR%? what’s means?

Line 168, Line 188: p italic

Figure 1: The ABCD of all figures should be consistent.

Line 493,500: Acronyms/Abbreviations/Initialisms should be defined the first time they appear in each of three sections: the abstract; the main text; the first figure or table. When defined for the first time, the acronym/abbreviation/initialism should be added in parentheses after the written-out form. (WCC)。 

In addition to the above issues, there are many details that need to be modified. Please revise carefully. The format should be unified in the full text. Part of them have been highlighted. Please check them throughout the paper.

The English language was difficult to follow in some sections of the manuscript; please carefully revise the details of the entire manuscript.

Author Response

Dear Reviewer,

We would like to express our gratitude for taking the time to review our manuscript titled "Monilinia fructicola response to white light", submitted to Journal of Fungi for consideration earlier this august. Your insightful comments and quick feedback have been a great help in refining the quality of the manuscript. I appreciate the thoroughness with which you assessed the work, and I am pleased to submit this revised version of the manuscript in response to your comments.

We have addressed each of your comments and suggestions point by point, and we would like to provide a detailed summary of the changes made:

  • The authors should check the use of capital letters and italics when referring to genes. Such as 22 or 27.
    • It has been corrected and revised throughout the whole manuscript.

  • Line 31-32: Acronyms/Abbreviations/Initialisms should be defined the first time they appear in each of three sections: the abstract; the main text; the first figure or table. When defined for the first time, the acronym/abbreviation/initialism should be added in parentheses after the written-out form.
    • It has been corrected and revised throughout the whole manuscript.

  • Line 47: When defined for the first time, the acronym/abbreviation/initialism should be added in parentheses after the written-out form. (RH)
    • It has been corrected.

  • Line 61-63: Space required before SI unit. (nm)
    • Spaces have been added before all SI units.

  • Line 66: Acronyms/Abbreviations/Initialisms should be defined the first time they appear in each of three sections: the abstract; the main text; the first figure or table. When defined for the first time, the acronym/abbreviation/initialism should be added in parentheses after the written-out form. (B. cinerea change to Botrytis cinerea). Such as Line 246,249,460
    • It has been corrected and revised throughout the whole manuscript.

  • Line 95: ml change to mL
    • It has been changed.

  • Line 114-115: 2.5 h 5 h change to 2.5 h, 5 h
    • It has been corrected.

  • Line 125: ml change to mL
    • It has been changed.

  • Line 129: HR%? what’s means?
    • That acronym is now properly explained in line 46-47 of the manuscript

  • Line 168, Line 188: p italic
    • It has been corrected and revised throughout the whole manuscript.

  • Figure 1: The ABCD of all figures should be consistent.
    • Captions are now consistent throughout all the figures

  • Line 493,500: Acronyms/Abbreviations/Initialisms should be defined the first time they appear in each of three sections: the abstract; the main text; the first figure or table. When defined for the first time, the acronym/abbreviation/initialism should be added in parentheses after the written-out form. (WCC).
    • It has been corrected and revised throughout the whole manuscript.

In addition to this, we have revised carefully the whole manuscript and made changes in wording, use of English and formatting of the text, in order to improve overall clarity and readability of the manuscritpt.

We believe that these revisions have significantly improved the manuscript, aligning it more closely with the standards of Journal of Fungi

Please find attached the revised manuscript along with a marked-up version highlighting the changes for your convenience. We hope that you will find the revisions to be satisfactory.

Thank you for your time and consideration.

Sincerely,

Dr. Antonieta De Cal

This manuscript is a resubmission of an earlier submission. The following is a list of the peer review reports and author responses from that submission.

Round 1

Reviewer 1 Report

Light responses of fungi have been extensively studied. Effects of light on development and virulence were widely reported in fungal pathogens, including in fruit postharvest pathogens. The present study investigated the light responses of Monilinia fructicola. In general, most results in the study are descriptive and not exciting, except the identification of far1. Unfortunately, they did not carry out a further study on the gene because of the lack of a transformation technical system. Gene knockout mutants of M. fructicola have been successfully constructed in the following study for your reference. Though the authors have pointed out this limitation in the discussion, the manuscript does not reach the publication standard of JoF.

CHEN  S  N,  YUAN  N  N,  SCHNABEL  G,  et  al.  Function  of  the genetic  element   ‘Mona ’  associated  with  fungicide  resistance  in Monilinia fructicola[J]. Mol Plant Pathol, 2017, 18(1): 90-97.

Reviewer 2 Report

Dear authors,

The manuscript “Monilinia fructicola response to white light” by Astacio et al. is well written and the data are presented in a clear way. The manuscript reads well and only small editorial changes to the language are needed. I will try to highlight some whenever possible, but as there are no line numbers, this is a bit difficult and there will probably be more. Please check the manuscript over again.

Here are some concerns:

I’m a bit concerned/confused about the data and the number of repetitions for all figures. It is stated as “it was repeated twice” in most figures. In biological science it is usual to repeat the experiments 3 times as independent biological replicates and on top of this sometimes with technical replicates. Otherwise, statistics are not really possible or meaningful. How did the authors design the experiments? For the shown growth assays in figure 2 for example: did the authors in experiment 1 use the same inoculum and put it on 5 different plates (technical replicates) or did the author use 5 independent, different inoculums to put it on 5 plates? And this was repeated a second time? Were the data of technical replicates averaged first to obtain the average of the first biological average and afterwards the biological repeats were averaged? (If, it was 5 biological replicates as stated in M&M that would be more than enough and a second full experiment would not be necessary, but would be fine if the authors chose to do so. But this should then be stated in the figure legends as 5 biological repeats were performed and the whole experiment was repeated a second time (then you should have 10 biological independent measurements for each strain?!?))  

Is standard error or standard deviation shown? Was the statistic test an ANOVA with Tukey’s on the averaged biological values? Please use for all figures standard error or standard deviation, but don’t switch between those in different graphs or subgraphs. If the RT-PCR graphs are standard deviation, then it should not be a problem to have standard deviation for the growth assays.

Among the 165 genes from 38 that showed modified expression after 5 min,…. Did you perform a gene enrichment analysis such as GO term for the RNAseq data for the 5 min and 24 h data? Or even for the overlap between the 5 min and 24 h data? This could/should be included here. Also, the Raw data for the RNAseq should be deposited in a public data base and the deposition number should be named here. Thus, the research community can reassess the RNAseq if needed.              

Are the gene numbers actually protein numbers? When I used the number for far1 at NCBI, it only worked for the protein database not the gene database. Please clarify, distinguish between gene and protein.

Figure 5 and 6 do not include statistics and again mention 2 separate experiments. Is it the same multiple 3-5 biological repeats and then the whole experiment was repeated a second time? Figure label should be “gene expression change (log2)” the deltadeltaCt is mentioned in the legend, which is good enough.  

Discussion:

The authors totally ignore to discuss the activation of different photochroms by different wave length of light. The readers don’t know the exact composition of the white light used by the author’s. But for sure the composition of the different used artificial lights is not similar to the solar light composition. For example, if the used white light (840) for the RNAseq did not contain high levels of blue light the white collar complex would not be activated and consequently there would not be a change in gene expression of white collar controlled genes. The same holds true for red or far red wave length. I don’t say the white light approach was wrong as long as the used light source is named and the photon emission rate is known, but there are now few examples for fungi which were treated with different exact wave length of light with an exact dose of photons and the gene expression changes regarding for example blue, red or far-red light have been published. The author should discuss this and put their results in context and even caution the irradiation with a specific wavelength of light might change the expression of other genes in M. fruticola compared to their used white light.

Just a quick look for Osram light bulbs light spectrum compared to the sun light spectrum already shows the differences:

Indeed the 840 light bulb used for the gene expression/RNAseq experiments is rather low in blue light.

Maybe it would be a good idea to determine the light wave composition of the different used light sources and photon flow rate/intensity at the used distances for the experiments and deposit it as supplemental data.   

Which light source was used for the infection test for RNA isolation and consequent gene expression analysis?

Abstract:

Change: filum to phylum

 M&M:

scratched to scraped?

Lights were turn on: turned on? Throughout

Figure 1 effect of white light?

Results:

Clarify/rewrite the paragraph it is not clear what is meant: Since photoperiods are important on fungal growth,…….     

Figure 2 and all following figures: the Y axes should be labeled with a clearer description of what is shown such as “radial growth in mm d-1” or “formed conidia per mm per day as log10”. Also the a,b,c,d to indicate sub-graphs should not be close to any bar if the stats are shown as a,b,c. in graph A) I first thought the a in the top right corner is a significance symbol, Maybe use A,B,C,D and put t in the top left corner?

Table 1. related proteins present in,….  B05.10 not B05,10

-proteins with the LOV domain but any output domain. Not clear what is meant

- conidial suspension and which after 24h incubation in darkness were exposed or not to light. Not clear

The manuscript reads well and only small editorial changes to the language are needed. I will try to highlight some whenever possible, but as there are no line numbers, this is a bit difficult and there will probably be more. Please check the manuscript over again.